# Nutritional Considerations of Irish Performance Dietitians and Nutritionists in Concussion Injury Management

**DOI:** 10.3390/nu16040497

**Published:** 2024-02-09

**Authors:** Emma Finnegan, Ed Daly, Lisa Ryan

**Affiliations:** Department of Sport, Exercise and Nutrition, Atlantic Technological University (ATU), H91 T8NW Galway, Ireland; emma.finnegan@atu.ie (E.F.); ed.daly@atu.ie (E.D.)

**Keywords:** concussion, mild traumatic brain injury, nutrition, sport, recovery, dietary, supplement, athletes

## Abstract

Sport-related concussion incidence has increased in many team-based sports, such as rugby, Gaelic (camogie, hurling, football), and hockey. Concussion disrupts athletes’ brain function, causing an “energy crisis” that requires energy and nutrient support to restore function and heal. Performance dietitians and nutritionists play a role in supporting athletes’ post-injury nutritional demands. This study aimed to investigate Irish performance dietitians’ and nutritionists’ knowledge and implementation of nutritional strategies to manage and support athletes’ recovery following concussion. In-depth, semi-structured interviews were conducted with seventeen (n = 17) Irish performance dietitians and nutritionists recruited from the Sport and Exercise Nutrition register and other sporting body networks across Ireland. Participants practised or had practised with amateur and/or professional athletes within the last ten years. All interviews and their transcripts were thematically analysed to extract relevant insights. These data provided valuable insights revealing performance dietitians and nutritionists: (1) their awareness of concussion events and (2) their use of nutritional supports for concussion management. Furthermore, the research highlighted their implementation of ‘novel nutritional protocols’ specifically designed to support and manage athletes’ concussion recovery. There was a clear contrast between participants who had an awareness and knowledge of the importance of nutrition for brain recovery after sport-related concussion(s) and those who did not. Participants presenting with a practical understanding mentioned re-emphasising certain foods and supplements they were already recommending to athletes in the event of a concussion. Performance dietitians and nutritionists were keeping up to date with nutrition research on concussions, but limited evidence has prevented them from implementing protocols in practice. Meanwhile, participants mentioned trialling/recommending nutritional protocols, such as carbohydrate reloading, reducing omega-6 intake, and acutely supplementing creatine, omega-3 fish oils high in Docosahexaenoic acid, and probiotics to support brain healing. Performance dietitians’ and nutritionists’ use of nutrition protocols with athletes following concussion was linked to their knowledge and the limited scientific evidence available. Nutrition implementation, therefore, may be overlooked or implemented with uncertainty, which could negatively affect athletes’ recovery following sports-related concussions.

## 1. Introduction

Sport-related concussions (SRCs), also known as mild Traumatic Brain Injuries (mTBIs), are prevalent [1,2,3,4], with up to 3.8 million cases reported in the United States annually [5,6,7]. The term ‘mild’ does not reflect severity but reflects the absence of structural brain damage. SRC is a type of mTBI with subtle differences. Both (SRC and mTBI) share subjective presentations [8,9] and are often used interchangeably, particularly in the sporting context [2]. Improvements in technology, education, and concussion identification protocols [10] have led to an increase in the reporting of SRC incidents [11,12], though many are still going unreported or undiagnosed [13,14].

SRCs are inevitable in contact and collision sports. Due to their subjective and invisible injury presentation, many frequently go undiagnosed or unreported [2,3,4,15,16]. In sports, sub-concussive impacts, and repetitive head injuries (RHIs) are concerns due to their long-term effects on athletes’ health [17]. In Ireland, many athletes (amateur and professional) engage in collision sports, contact sports, or other activities that will increase their exposure to SRC events [18]. Collision sports, such as rugby union, involve body-to-body collisions as a fundamental element of the game. For contact sports, such as Gaelic football, hurling, camogie, field hockey, and soccer, purposeful body-to-body collisions or outright tackling [19], although not endorsed, are recognised as a natural part of the game [20,21]. Combat sports involve striking (boxing), grappling (wrestling), or a combination (mixed martial arts) [22,23] and involve ‘knock out’ or a ‘technical knockout’ for a fight to cease [24]. Additionally, non-field-based activities such as cycling can involve bike collisions and crashing [25]. Concussion does not discriminate based on age, gender, preferred sport/activity, or level (amateur/professional) and impacts individuals of all backgrounds [26,27,28,29]. In Ireland, athletes will often play multiple team sports recreationally, at an amateur level (Gaelic, combat, etc.), which typically have limited medical field side support. This situation increases the risk of SRCs going undetected or undiagnosed and athletes’ exposure to RHIs [10]. Scullion et al. [30] identified that players and coaches in amateur sports may have limited knowledge and training regarding SRC, in contrast to those in professional or elite team settings [30]. Therefore, providing suitable education, training, and resources to athletes, coaches, and medical professionals would help reduce harmful outcomes and long-term consequences associated with SRC [31].

SRC, according to the “Concussion In Sport Group” (CISG), is provoked by a direct or indirect impact to the head, face, neck, or other body parts, passing an impulsive force to the brain, disrupting its function, and resulting in the rapid onset of transient symptoms [1,9]. Symptoms and signs are subjective, and present immediately or develop over minutes or hours, altering athletes’ cognitive and mental state (e.g., confusion, disorientation) [1,9,30,32]. Athletes present with SRC in a variety of ways and do not have to have visible head trauma or a loss of consciousness. SRC, if experienced in isolation and managed properly, should not pose a significant risk to athletes’ health. Most individuals should see improvements in their symptoms within 7 to 30 days [3,17,33] and return to their baseline within 3 months [34] following concussion. Recent research has highlighted the potential importance of nutrition in supporting and promoting recovery from brain injury, making it an essential component currently understated for SRC reduction and management [3].

Though an emerging area, in the concussion recovery management literature, nutrition is acknowledged for providing the fuel and nutrients to meet functional healing and repair demands [3,11]. In some instances, athletes have insufficient stores of specific nutrients (such as vitamin D) for neuroprotection and acute healing and recovery support. At present, there are no clear recommendations on strategies and recommended dosing [35,36] post-SRC. Strategies to increase athletes’ intake of protein foods would increase amino acid availability and may prevent further tissue breakdown, support healing, and regulate brain metabolism [3,37]. Increasing athletes’ consumption of fats, particularly omega-3 fatty acids such as docosahexaenoic acid (DHA) and eicosapentaenoic acid (EPA), through food (fish and algae-based products) and supplements may also support recovery. DHA plays a role in brain development [38] and has been shown to be neuroprotective, having and may aid brain recovery following RHI and concussion [3,39]. Supplementation with a high dose of DHA or a combination of DHA and EPA may help saturate brain cells and promote healing [3], mitigating the long-term adverse effects associated with low brain DHA levels after concussion [40]. No specific DHA recommendations exist for high-risk sub-concussive sports athletes [41]. However, the Food and Drug Administration (FDA) and the European Food Safety Authority (EFSA) recommend consuming at least two servings of fish per week and a maximum of 2–3 g daily via supplement (EPA and DHA) for optimal health [3,42,43,44]. In addition, the fat-soluble nutrient vitamin D is neuroprotective and promotes healing and neural recovery following a head impact [3]. Athletes with low vitamin D levels have reduced neuroprotection, heightened inflammation, and compromised and prolonged cognitive recovery [3,45]. Supplementing magnesium within 24 h replenishes depleted brain cellular levels, alleviating symptoms and promoting faster recovery [46].

Creatine, an essential amino acid, is crucial in energy production, influencing brain and muscle tissue function and performance, and may play a role in brain injury recovery [47,48]. Without free creatine or phosphocreatine available, energy deprivation occurs during the neurometabolic cascade following the accelerated uptake and use of adenosine triphosphate (ATP). Increasing cellular creatine levels in the hypometabolic period may help restore energy by supporting ATP production from glucose and enhancing oxygen supply to the brain. However, positive effects of creatine have been primarily observed in healthy individuals, animal populations, or cases of severe brain injury. Its effectiveness in addressing concussion-related (RHI or mTBI) issues is uncertain and requires further investigation for preventive and post-SRC protocols.

The field of performance nutrition is dynamic and involves applying evidence-based nutritional strategies to help enhance athletes’ training, recovery, and athletic capacities [49]. In the event of an injury, performance dietitians (PDs) and nutritionists (PNs) help athletes in meeting their nutritional needs to support healing and recovery using dietary and supplementation strategies. Given their pivotal role in supporting athletes after injury, this research aimed to understand the nutritional protocols Irish PDs and PNs implement with their athletes post-concussion.

## 2. Materials and Methods

### 2.1. Study Design

Braun and Clarke’s reflexive thematic analysis (TA) approach was implemented to enable the interpretation of the qualitative data to identify and analyse themes and patterns present throughout the research data [50]. Individual semi-structured interviews were designed, conducted, and reported in line with the COnsolidated Criteria for REporting Qualitative research (COREQ) [51].

Before the study commenced, interview guides were developed by E.F. and piloted with two PNs to test the feasibility of the interview script questions. All participants in this study were performance/sports dietetic and nutrition professionals (female and male) located and practising in Ireland or Northern Ireland and working with professional/elite, semi-professional/sub-elite, or amateur athletes and/or teams aged ≥16 years. Professional registrations (the Sport and Exercise Nutrition register (SENr)), sporting bodies, the Irish Rugby Football Union (IRFU), the Gaelic Athletic Association (GAA), Sport Ireland, and sports nutrition networks were used to identify and source PN(s) and/or PD(s) that were currently practising throughout Ireland/had practised within the last ten years. The lead researcher (E.F.) invited suitable candidates to participate in an online one-to-one interview via email.

Data collection began in March 2022 and ceased in August 2022 once data saturation had occurred. The interview format was designed to discern participants’ backgrounds, education, and career paths to date; the different types of athletes they worked with; sports; and settings. Section B then proceeded to ask participants about the types of injury their athletes would commonly present with and if, as practitioners, they considered/used nutrition to support athletes’ healing and recovery outcomes. The final section sought to identify specific nutrition protocols PD(s)/PN(s) implemented for different types of injury, tissues, and sites to support and facilitate athletes in the recovery and healing process.

### 2.2. Ethics and Procedure

This study was ethically approved by the Research Sub-Committee of Atlantic Technological University’s Academic Council (ATU-RSC_AC_25/11/2021). The lead researcher (E.F.) identified the initial number of participants by distributing a direct invite alongside a participant information sheet detailing the purpose of the study. Before each interview, informal discussions were conducted with participants to discuss the study objectives, protocols, and the confidential management of the resulting data. The lead researcher (E.F.) conducted all interviews using Microsoft Teams, and before each interview, participant consent was recorded. Interviews lasted between 25 and 60 min (Mean ($\bar{x}$) = 41.77 min, Standard Deviation (SD) = 9.9 min) in duration. Interviews were semi-structured, with a topic guide that included a series of ethically approved open-ended questions and supporting questions designed to create discussion on topics relevant to the study. After each interview, each data transcript was collected, and all data related to the study were fully anonymised to protect all professional participants from identification.

### 2.3. Participants Characteristics

Seventeen (n = 17) PN(s) and/or PD(s), identifying as male (n = 8) and female (n = 9), working with a wide range of professional/elite, semi-professional/sub-elite, or amateur male and female (adult and youth) athletes took part in the interviews. During data collection, participants were practising in the Republic of Ireland and Northern Ireland or had been within the last ten years.

### 2.4. Sampling and Eligibility Criteria

A purposeful sampling [52,53] methodology was employed to create a sequence of responses across participants. An Irish network of PN(s) and PD(s) was identified, targeted, and contacted, with an invitation to participate in the one-to-one interviews. Professional registrations (SENr), sporting bodies (IRFU, GAA, and Sport Ireland) and sports nutrition networks were used to identify and source PN(s) and/or PD(s) registered to practise throughout Ireland currently (at the time of the study) or within the last ten years. Eligible participants had to be working with adults and youth (≥16 years of age), professional/elite, semi-professional/sub-elite, or amateur athlete(s) and/or teams. E.F. invited suitable candidates to participate in online one-to-one interviews via email. The sampling process was concluded once data saturation was reached.

### 2.5. Data Collection

Interview questions were used to gather evidence of PD(s)/PN(s) personal experiences with injuries, specifically concussions. They discussed awareness of injury events within their athlete cohorts, their role in prevention and injury management, and how they utilised nutrition to support athletes’ return to play (RTP) after concussion.

### 2.6. Researcher Description

The lead researcher and author (E.F.) identifies as a woman and has been actively involved in sport and nutrition professionally for eight years in consultancy and research settings, working with athletes and performance nutritionists in amateur, semi-professional, and professional sports settings. E.F. is currently a research associate within the Irish Concussion Research Centre (ICRC) engaged in researching strategies for managing, reducing, and educating cohorts about mTBI/concussion, RHI, brain health, and nutrition. Her research focuses on investigating the “Role of Nutrition in Concussion Recovery and Management”. During the research process, to increase rigour, E.F. implemented a bracketing method to support their background reflection before data collection commenced to minimise potential bias from personal preconceptions [54].

### 2.7. Data Analysis

A reflexive TA approach was implemented due to the researcher’s role in knowledge production [50]. Reflexive TA would support a richer and more precise reading of the data as it required the researchers to continually question and query any assumptions made during the interpretation and coding of the data [50]. The analysis followed a six-stage approach to identify and construct meaning and patterns from the data [50,55].

### 2.8. Transcription

Initially, the original audio recordings of each interview were listened to multiple times before being transcribed verbatim by the lead researcher (E.F.). Transcripts were examined for accuracy, cross-checked with original audio recordings, and re-edited as necessary. This familiarisation (Phase 1) with the content of each transcript was continued by E.F., who re-read each transcript on four separate occasions and compiled initial notes per interviewee to support the coding process. This practice enabled E.F. to confirm that questions and responses were linked with each interview audio recording. Transcripts were then read by a second author (L.R.) blinded to the participants and any initial trends generated within the data.

### 2.9. Coding

Initial codes (Phase 2) were created based on theoretical assumptions, and the research aimed to focus on semantic meanings within interview transcripts. This was carried out by E.F. and reviewed together with L.R. Both researchers independently and systematically coded transcripts and then met to discuss and verify codes and their application to research data. This coding phase created theme semantic blocks (Phase 3) within transcripts, which were then linked to subthemes, and quotations were extracted (Appendix A) accordingly. The authors reviewed themes (T) in Phase 4 to check and challenge any assumptions. Thematic maps (Appendix A) were developed, and theme linkages were created and reviewed to convey meanings evident throughout the dataset. The initial Ts included (T1) Concussion Awareness—reporting and communication, (T2) Nutritional Support—Awareness, (T3) Knowledge of Nutrition Protocols, (T4) Nutrition Implementation, and (T5) Nutrition Novel Protocols—specific to concussion. Themes were defined and refined (Phase 5) to ensure they individually captured each theme clearly. As a result, theme 1 was renamed, to be more specific, and themes 2, 3, and 4 were joined (Figure 1A,B). The final themes were PDs’/PNs’ ‘(T1) Awareness of Concussion Events’ and their use of ‘(T2) Nutritional Support for Concussion Management’, which included ‘Novel Nutritional Protocols’ to support and manage SRC recovery. For the final stage (Phase 6), an analytic narrative was produced, and this is presented in the Results section (Section 3) in this manuscript.

## 3. Results

The purpose of this study was to develop an understanding of the nutritional protocols implemented by Irish PDs and PNs with their athletes post-concussion. The semi-structured interviews TA identified two main themes (T) with eleven subthemes (s-T). Descriptions of each central theme (T = 2) and its associated subthemes (s-T = 11) are summarised in this section (3.0) with meaningful verbatim quotation evidence to highlight participants’ narratives.

### 3.1. Participant Characteristics

Participants (Table 1) were Irish PDs and PNs working with a wide range of adult and youth (≥16 years) professional/elite, semi-professional/sub-elite, or amateur athletes across the Republic of Ireland and Northern Ireland. At a minimum, participants held Master of Science degrees in sports nutrition and exercise and/or a Doctor of Philosophy (PhD). The PDs and PNs involved in supporting team-based athletes were participating in various Irish native sports, including Gaelic football (https://www.gaa.ie/my-gaa/getting-involved/gaelic-football (accessed on 21 June 2023)), ladies’ Gaelic football (LGFA, https://www.gaa.ie/my-gaa/getting-involved/ladies-gaelic-football (accessed on 25 June 2023)), hurling (https://www.gaa.ie/article/hurling (accessed on 25 June 2023)), camogie (https://www.gaa.ie/my-gaa/getting-involved/camogie (accessed on 25 June 2023)) (intercounty, county, and club levels), soccer (Premier League, League of Ireland, and club-level), rugby union, women’s hockey (provincial and club-level), and cricket (professional). Individual sports were also represented, such as combat sports, boxing, mixed martial arts, kickboxing, and judo (amateur and professional/elite levels); gymnastics (elite); athletics, running, and cycling (elite, ex-elite, and amateur levels); rowing; endurance events; triathlon (Ironman); and competitive multisport, Olympic lifting, and powerlifting.

### 3.2. Thematic Results

Two major themes were identified from this research analysis of PDs and PNs: (1) their awareness of concussion events and their use of (2) nutritional supports for concussion management (Figure 2), including ‘novel nutritional protocols’, to manage SRC.

These main themes were separated into subthemes, theme 1 and 2 (Table 2 and Table 3).

#### 3.2.1. Irish Performance Dietitians and Nutritionists’ Awareness of Concussion

##### Concussion: Personal Experience

In the context of concussion awareness, participants who cited having prior personal experience diagnosing or reporting concussions exhibited a higher awareness of their impact on athletes. In comparison, those without such experience demonstrated a lower awareness. The participants reporting concussion as common, casual occurrence amongst their athletes showed higher awareness of injury incidence without prompting or hesitation.

“*I suppose obviously within… within rugby; we have like a lot of the collision-based injuries. So, for example, you know, the likes of concussion come up a lot and kind of ligament… ligament and tendons would be the ones that I see a lot of…*”[PN 17]

In 2014, World Rugby introduced the Head Injury Assessment (HIA) concussion management system into an injury policy [56]. One participant mentioned that head injury such as concussion “*wasn’t really like a… like a thing… before HIAs*” [PN 3]. Before this, PDs/PNs involved in rugby were not part of the conversation or made aware of concussion incidence compared to 2022, when HIA was protocol and actively used to assist and improve reporting practices. However, in PDs/PNs working with hockey, soccer, combat, Gaelic football, and some individual athletes, reporting was not evident as they showed no awareness of concussion occurrence as an injury subset. In this instance, PDs/PNs reported that they were fortunate to have no recorded cases of concussion among their athletes:

“*…I’m so lucky that, in just having the conversation with you there now… I’m just thinking back like five years, five years working in performance nutrition directly, and I have never had anyone with real…real acute injuries like*.”[PN 10]

Alternatively, it is possible that they may be unfortunate, and that in reality, underreporting is going unnoticed. Another participant highlighted the prevalence of injuries among high-level combat athletes and pointed out that “*…you never even hear of like concussions reported back to you, but God knows, even though it’s probably likely you’ll never hear back!*” [PN 12]. Meanwhile, a PN with a background as a hockey player shared insight into their injury experience, having had “*…three concussions at one point in 18 months*” [PN 11], highlighting differing levels of cognitive decline experienced and their subsequent need for nutrition to fuel their healing and extra rest. However, the PN was unaware of concussion incidences within their own hockey or soccer athletes (see Table 2).

##### Concussion Communication

The interviews generated patterns linking PDs’/PNs’ experiences and awareness of concussion to the level of communication that existed between back-team members and their athletes in the incident of injury. Delving deeper, the data highlighted incidences of poor communication where the PDs/PNs were overlooked and, as result, did not know athletes had sustained an impact or SRC. PDs/PNs admitted feeling the need to be present at training and matches to be aware and in the know regarding SRC events. They felt the onus was on them as practitioners; otherwise, their involvement/awareness of injury events would be easily overlooked, especially if they were not present.

“*Luckily, we don’t really get too many of them now like that I can; I can miss some as well just by not being on site too often…*” [PN 4]

Having an awareness and understanding of an athlete’s health and injury status, specifically after concussion, is a multifaceted task that requires collaboration across the entire back team. It is not the sole responsibility of one professional (i.e., doctor or the physio) in isolation to provide support, as highlighted by PN 4:

“*I know when there is a decent backroom team that are all together, and I know when…they don’t really believe in all the different pillars as much as their own pillar… If you’ve got a backroom team that is set up in a very efficient way, it’s just them firing off a message to one person who does their part…*”[PN 4]

PDs/PNs know that differing communication and collaboration levels are present within a sport setting and are aware of their impact on forming “*a decent backroom team*” [PN 4]. A well-coordinated backroom team characterised by shared values, mutual respect, and effective communication collaborating on members’ expertise would guarantee the successful implementation of essential measures following an SRC.

While the ideal scenario involves strong collaboration within the back-team setup, it is not always common practice, as expressed by PN 4 sharing the impact of limited collaboration situations where they felt it had not “*…been as strong, you know, and I think that makes a big difference in terms of catching the acute phase*” [PN 4]. In addition, PN 13 commented on the value placed on the PDs’/PNs’ discipline and noted that at

*“…the amateur, intercounty level…nutritionists are nearly shoved to the side ‘ahhh we don’t need the nutritionist on match day’, you know, but like physios they’re stade on foot…it’s happening at all corners, and that’s at the top level*”[PN 13]

This highlighted the potential dangers of scenarios where the back-team setup is not valued, is fragmented, and has poor communication channels. In contrast, when communication channels are open, PDs/PNs have injury awareness. PDs/PNs can effectively prepare and tailor their support and work with back-team members to ensure athletes’ best recovery interests are met, emphasising the importance of a cohesive relationship and collaborative approach. Furthermore, providing more insight into the dynamics of the back team, PN 5 stated:

*“…if you know the doc is on to them if you know the coach, the manager is on to them… if you know the physios are on to them, it sometimes best to void your time… I’ve gotten a little bit better at identifying when you are helpful and when you are just annoying them, you know?*”[PN 5]

For some participants, their first protocol following a suspected SRC impact is a diagnosis and medical sign-off; they know this is a crucial step before any other back-team member implements support: “*on the day I would be leaving it to the medic, really*” [PN 1]. However, PDs/PNs seemingly underestimate the significance of their role in acute recovery support compared to other back-team members.

“*I think in that scenario (pause)… I would wait to hear from…some other medical professional before doing anything…**Rugby is one of those sports…I think that it’s further ahead than a lot of sports certainly in Ireland, where they do have that support on hand.*”[PN 16]

##### Concussion Attitudes

Awareness of concussion incidence amongst PDs/PNs appears to be influenced by a seemingly ‘blasé’ attitude toward the injury. This may influence an athlete or backroom team member’s reaction and subsequent reporting behaviour if a suspected concussive impact occurs. PDs/PNs working with athletes in contact collision-based sports, like rugby, expect injuries due to the nature of the sport and perceive athletes to often be in a heightened recovery state, especially after matches. For example, PN 8 commented:

“*Overall, there’s nearly always have some kind of knock or a niggle or a strain… When we’re coming out off the back of matches…**They’re just constantly in a state of basically recovery because they come out so beat and bruised from games.*”[PN 8]

However, could this better awareness be creating a more casual attitude toward concussion, as some participants comment and perceive it as “*just another like…injury*” [PN 8]?

“*…with guys that are there concussed, which are probably be quite regularly… which is unfortunately the case, your kind of on to them to stay away…*”[PN 2]

Athletes in collision sports may have a casual attitude due to the subjective nature of concussions. For instance, they may believe that they can quickly RTP, as stated in an example of an athlete persona by PN 8: “*…ahhh no I’ll be fine, I’ll be back in a week*” [PN 8]. The frequency of SRC in a collision sport and its subjective presentation (i.e., symptoms) may feed behaviours of poor compliance to recovery support programs. As stated by one PN, speaking specifically about professional rugby athletes,

“*They don’t really care… I think it’s cause it’s such a common injury, like while concussion in GAA might be less common. When it’s so common in rugby…concussion is just like another…another injury*”[PN 8]

Athletes tend to disregard subjective and invisible injuries, especially when they are eager to RTP. As noted by PN 7, athletes can be “*…very quick to dismiss…injuries, especially when, I suppose, there’s no, not that there’s no obvious symptoms of it…you know, they’re not carrying their leg*” [PN 7] due to physical symptoms such as limping or bruising. This behaviour is commonly observed among younger rugby and Gaelic athletes, who may be more inclined to take risks to get back on the field, as mentioned both by PN 7 and PN 8.

##### Concussion Awareness Evidence

In elite-level combat sports, where concussions are frequently reported, some PDs/PNs proactively implement objective tests and diagnostic protocols to help athletes better understand the impact of SRC on their brains. One such protocol involved using the ’Sport Concussion Assessment Tool 5 (SCAT5)’ [57,58] to assess brain function and provide athletes with objective evidence (subjective) of injury and concussion. In addition, PN 14 explains that SCAT5 results help demonstrate the significance of proper hydration to an athlete on their brain function and may support concussion prevention, given that “*if they’re dehydrated, their response to the SCAT5 is actually a lot less*” [PN 14].

Several participants showed awareness and knowledge of the acute metabolic and physiological changes that occur in the brain tissue following a concussion, which reduces energy availability, mentioning the following statements:

“*But like, even if you look at concussions, like concussions, like a shortage of energy in the brain at the time of impact*” [PN 9] (referring to increasing energy/caloric intake).

Additionally, in the period following the concussion,

*“to help with that energy availability side of things… Ehmm, if the brain can’t fully use their glucose efficiently”* [PN 8] (referring to implementing creatine supplementation).

These statements reflect PDs’/PNs’ genuine understanding and knowledge of the manifestation of concussion as an injury and the associated metabolic and functional changes in the brain tissues’ physiology.

#### 3.2.2. Theme 2: Nutritional Support for Concussion Management

##### Awareness of Using Nutrition in Concussion Management

Participants’ consideration of nutritional strategies to support their athletes with SRC was linked closely to their awareness and understanding of nutrition for its role in healing the brain tissue during recovery. This was evidently reflected through PDs’/PNs’ implementation of nutritional support protocols for athletes with SRC.

However, participants had mixed levels of awareness and knowledge; some presented uncertainty around SRC as an injury and its subsequent need for nutrition, impacting their knowledge of strategies to support and manage recovery. PDs’/PNs’ lack of awareness and nutrition knowledge was a concern as they were not well informed about its role following SRC. As a result, they did not see or consider the benefits of using nutritional support to promote brain tissue healing, which could pose a risk to their athletes.

“*Not too sure, really, just keep up the good habits. But I haven’t looked at it… Is there much? Is there much research on concussion and nutrition immediate or…*?”[PN 1]

A link was identified between participants who were unaware of concussion incidence within their team setup and/or athletes. These participants did not consider and were unaware of any physiological and metabolic changes that happen in the brain after impact, providing vague responses revealing a limited understanding. In addition, they had poor knowledge of the significance of adjusting athletes’ nutritional intake (for perceived ‘invisible injury’) to meet altered requirements to facilitate healing and recovery post-SRC.

For example, PN 15 commented,

“*I think it’s a really good way of putting it in terms of like an invisible injury. You don’t, you don’t necessarily think that there’s a, there’s a change required. I guess they’re not wrong in saying that there probably isn’t from a nutrient point of view.*”[PN 15]

A number of PDs/PNs were aware of concussion events and the role of nutrition in brain health as they actively kept up to date with scientific research. However, informed participants were not confident or certain on nutrition protocols and considered strategies based on their acquired knowledge influenced by limited scientific evidence.

“*I know there is a little bit on creatine and omega-3 fish oils as well. Ehmm, so again, not a whole lot we can do.*”[PN 7]

Participants who demonstrated a more certain type of knowledge actively kept informed by scientific research related to concussions/brain injuries and the role of nutrition in brain health and function. However, they were unsure of protocols as they, too, were influenced and limited by “*no real hard and fast evidence on it*” [PN 9].

“*Yeah, it’s, it’s an area I’m doing a bit of reading around at moment. But definitely, concussions are not that well-researched. But I think from what I have read there’s a couple things you can do and creatine, I believe is important in the… not even the acute phase of, of, of having a concussion, but I think consistently for optimal cognition.*”[PN 2]

PDs’/PNs’ personal experiences with concussion and head injuries provided them with experiential knowledge on the importance of a nutritious dietary intake and regular eating habits for brain recovery. This experience led them to logically consider implementing nutrition strategies to support their athletes’ recovery following SRC prospectively.

“*So, what would I implement with somebody else? I think to be honest…the rest and the regular eating are, are crucial and then…the extra probiotics or, ehmm you know, whether they make that much of a difference, I don’t really know. But the anti-inflammatory, the protein, the carb, I think they would be crucial. Obviously, you know…your healthy fats as well. But the diet…the diet itself is probably the most important thing*.”[PN 11]

Some participants who lacked knowledge of potential SRC management protocols were not negatively limited as they had access to a network of expert nutritionists. This network helped bridge their knowledge gap and supported them implement necessary nutrition protocols with their athletes in the event of an acute injury.

“*With regards to concussion, then we have two dietitians. One that actually has worked in rugby and who would I suppose work and are more familiar with the concussion side of things…*”[PN 13]

This evidence presents an opportunity for PD/PN professionals who can often work independently to have an open network to share expertise and support the expansion of nutrition knowledge. The PDs/PNs who mentioned being able to refer to a niche network of nutrition experts with experience in SRC, as a result, had the support they needed to implement safe nutritional protocols to manage their athletes’ recovery. Interestingly, PN 9, despite having extensive knowledge on concussion and the sequel metabolic brain changes that occur, mentioned only implementing nutritional strategies at an individual level with athletes, unlike with other protocols formulated into their injury (rugby) management guidelines and policy. Experts expressed hesitancy and discomfort in recommending nutritional protocols following a concussive impact (invisible brain injury). This reluctance stemmed from the variability in research, such as the lack of human-specific studies and numerous unanswered questions and knowledge gaps.

PN 9 highlighted these concerns:

“*But like, even if you look at concussions, like concussions, like shortage of energy in the brain at the time of impact…you could, like, I’d broadly say it to people, you know…’just to make sure you eat enough during this time, and kind of, you know’, were kind of very basic interventions. But nothing, I don’t think we ever…got to a stage where we were comfortable giving nutrition advice to someone who was concussed because there was stuff out there on creatine and stuff on the high dose DHA, but…we never formalised that to a policy*.”[PN 9]

This led the nutrition focus to remain on physical or visible type injuries, which may cause other sports (outside of rugby) to follow suit and, as a result, neglect nutrition as a strategy (or PDs/PNs role) for brain healing following SRC due to the subjective and non-physical presentation of the injuries (creating a potential implementation barrier).

##### Implementation—Nutrition Post-Concussion

Participants who did mention attempts to implement or actively trial specific nutritional strategies to facilitate athlete recovery following SRC did so with an element of uncertainty. Following a known SRC, some PDs/PNs took a comprehensive approach aiming to minimise athletes’ mental load and any potential challenges faced by athletes during their injury recovery. They addressed athletes’ nutrition requirements, overseeing fuel selection (i.e., carbohydrates and fats), quality (i.e., protein), and nutrient variety (i.e., micronutrients) and implementing specific supplements for optimal performance and to facilitate SRC recovery.

In the event of injury, an athlete’s focus needs to shift from performing at their very best to healing and recovery to support their return to sport (RTS) [1]. During this period, the athlete’s mood may fluctuate between motivated and capable to depleted, depressed, and unmotivated, which may impact their nutritional habits. For example, suppose athletes’ habits fall by the wayside. In that case, their level of engagement with the PD/PN may become disjointed, which may/will impact the level of nutrition support they receive, and subsequently their recovery will suffer.

“*…if the player isn’t overly motivated, they’ll get blanket ehmm…they get blanket attention. The motivated player will get more attention. The educated player will snap back into their habits…*”[PN 1]

Some PDs/PNs were mindful of this shift and took practical steps to implement nutrition strategies specific to their athlete’s needs that would minimise their mental load. These strategies focused on ensuring athletes ate enough food to fuel and nourish brain tissue healing and support them in feeling better overall.

“*So tried to ensure energy levels were kept up just through kind of liquid-based meals, small little snacks, nothing that was going to cause…stomach aggravation… So, it was more kind of in that, you know, couple days after…after the knock as opposed to long term*”[PN 7]

Strategies to facilitate acute energy requirements post-SRC need to be tailored and consider the varying (subjective) symptoms experienced by athletes, such as being “*quite nauseous*” and unable “*to eat a lot*” (i.e., nausea), through implementing “*smaller meals, more frequently, liquid*” [PN 7] to help simplify and support adherence. PNs/PDs focus on and manage athletes’ dietary changes to nourish and support brain tissue healing by encouraging and supporting athletes’ intake of “*anti-inflammatory foods…your fruits, fruit and veg, fish*” [PN 11] and promoting their intake of poly-unsaturated dietary fats via omega-3 rich foods and supplements. Dietary fat plays a significant role in providing fuel and nutrient support for the brain. However, following injury, specific types become exceedingly important, and it is advised to limit the consumption of “*pro-inflammatory omega-6s*” [PN 6] foods to prevent an imbalance of omega 3s to 6s. Proactive PDs/PNs are strategic and align nutrition support with their athletes’ mindset and recovery goals at the forefront whilst syncing with other back-team members: “*…initially we try to break down even with the physios; we’d break down their rehab into, like, phases*” [PN 8]. Participants know their role in injury support and that when implementing nutrition, their concussed athlete needs clear “*…smaller focuses instead of bombarding them all at once*” [PN 8]. This approach aims to keep the athlete accountable and focused during SRC recovery, making the process less daunting and challenging.

“*So, the initial being, their protection phase, for example… I try row in with that being with ‘this is your initial focus’ instead of trying to bombard them with loads. Saying, ‘this is what I really want to get out of this block’, and then we tried to do it in focus*.”[PN 8]

Furthermore, PDs/PNs provide flexible availability with open channels of communication for athletes to seek and receive support during SRC recovery through casual check-ins, “*corridor conversations*” [PN 5], WhatsApp messaging, and planned one-to-one sit-down meetings around training schedules, either “*early from a training session, or you meet them there beforehand*” [PN 5]. Open channels and connecting with athletes in casual environments are highly beneficial. They provide opportunities to establish trust and build rapport as some athletes may not be very forthcoming. For athletes who may not have as much access to in-person support, PDs/PNs provide “*Zoom conversations, sharing resources on the screen*” [PN 5], communication, and support that work around lifestyles and outside stressors that factor into athletes’ SRC recovery trajectory.

##### Potential Barriers

Athletes’ attitudes on SRC as an injury and their value on the importance of nutrition and its role in their RTP strongly dictate buy-in and protocol implementation. Athletes tend to place a higher value on more physical injuries. They are more “*…open to the fact that nutrition can play a role immediately*” [PN 17], creating an adherence motive, compared to the invisible presentation of head injuries and concussions. In assisting athletes’ SRC recovery outcomes, PDs/PNs prescribe daily supplements that are brain-supportive and tick all their nutritional gap boxes. However, the reality is they are unable to control behaviours, as one participant admitted, “*Well, see, I don’t know if that actually taken it!*” [PN 12]. Meanwhile, PN 5 stated empathetic know-how in the event of injury that “*most, or some of their*” athletes’ “*good habits to go out the window*” [PN 5], commenting that in such incidences, “*…ideally you want to be on to those lads*” to remind athletes to “*keep on top of the…omegas and the creatine*” [PN 5]. Participants stressed the need to remind athletes of the basics that they should be adhering to habitually and taking, for example, fish oils. However, PDs/PNs are cognisant that adherence is subjective and “*not everyone will take fish oils*” [PN 12]. Unlike measurable supplement adherence evidence,

“*…the guys who have been concussed recently… I definitely know one of them 100% was because he was looking for more creatine and everything, you know, because he was loading up on it*”[PN 12]

##### Novel Nutritional Protocol Implementation Post-Concussion

PDs and PNs mentioned exploring, considering, and trialling several novel nutrition protocols that may support the recovery of athletes who have suffered SRC. Numerous participants mentioned that their athletes had already been prescribed creatine for muscle mass and performance (i.e., strength) properties. For example, in rugby:

“*…a lot of our athletes would take creatine. But if they haven’t or around a concussion, potentially increasing that creatine intake*”[PN 8]

Creatine is considered in combat sports when there is a “known concussion in boxing…” [PN 15] to potentially help “where the boxer has to be pulled out, we have actually put in creatine to try and see if that can help in terms of the recovery from that” [PN 15]. One participant (n = 1/17) recommended a protocol implementing creatine supplementation with concussed Gaelic athletes. It entailed a loading phase, shifting athletes’ everyday prescribed performance dose from “5 g a day” to “20 g a day” [PN 12] following the SRC event (see Figure 3).

“*I’ll even tell them to load creatine and during that period…the week after a concussion, instead of just taking 5 g a day, I’ll get them to load like 20 g a day of creatine just for that little period…then the rest is kind of lead…by the physio*”[PN 12]

Participants cited using omega-3 fish oil supplementation with their athletes; some had no clear rationale, and others prescribed it for brain health, “*cognition*” [PN 2], and recovery properties. Following SRC, it was cited that athlete’s intake of “*healthy omega-3s*” [PN 6] should be increased and their “*omega-6s*” [PN 6] reduced. One PN (n = 1/17) recommended omega-3 supplementation following SRC and shared the protocol they implemented with Gaelic athletes. It included using a daily 3 g dose of an omega-3 fish oil to achieve a target ratio of 1.8 to 1.2, EPA:DHA, aiming to get athletes’ “*DHA levels up a little bit more*” [PN 12] (see Figure 3).

“*Ehmm, 3 about 3 g of…so that’s about 1.8 g. I think it’s like 1.8 to…the ratio was like 1.8 to 1.2, EPA:DHA…with the concussion. I’m trying to get the DHA levels up a little bit more.*”[PN 12]

The next protocol cited by one (n = 1/17) participant involved acute carbohydrate loading, which had been discovered from research findings indicating potential effectiveness in increasing energy availability in the brain after SRC, which may shorten symptom duration [59,60]. The protocol involved concussed collision-based sports (American football) athletes consuming 400–500 g of carbohydrates daily for 5 days during their RTP to meet post-SRC caloric requirements. The PN mentioned that they were trialling this protocol in their rugby setting by simply ensuring that athletes who had been concussed consumed enough carbohydrate-rich foods acutely following SRC.

“*…like 4 or 500 g of carbs in the 5 days, after concussion versus the guys that had a low carbohydrate diet and the symptoms for the guys that had more carbohydrates were got back to baseline that bit quicker.**So, that is kind of something I have been doing; it’s like, well, if a guy is in a gradual return to play protocol, then can you eat enough food? Specifically, carbohydrates…in that 5-day period. So that’s one new area…*”[PN 2]

PN 2 stressed the importance of ensuring athletes consume enough carbohydrate-rich foods post-SRC when implementing the protocol on a trial basis to potentially increase energy availability in the brain tissue and support overall recovery (see Figure 3).

The final nutritional protocol includes the use of a probiotic supplement; this was considered by one participant (n = 1/17) and came to fruition after PN 11 shared anecdotal experience after personally sustaining three concussions within 18 months. PN 11 highlighted how fuelling at regular intervals was essential to support their brain recovery and recommended that athletes supplement their food with a probiotic to help the gut–brain connection following concussion (see Figure 3).

“*…I would recommend a probiotic as well for a little while anyway… I know it’s not maybe an area that’s not that well researched yet, but the link between the gut and the head* (holding head).”[PN 11]

## 4. Discussion

The primary aim of the research was to understand the nutritional protocols implemented by Irish PDs and PNs with their athletes post-concussion to identify whether nutrition is being used to manage and support recovery following SRC diagnosis.

The interviews revealed that the levels of implementing nutrition practices among Irish PDs and PNs to support recovery following acute SRC diagnosis vary a lot. In the study, practitioners’ prior exposure to SRC greatly influenced their current knowledge, consideration, and use of nutrition as a support protocol in concussion management. A number of external factors heavily influenced participants’ awareness of concussion; these included other team members’ (athletes and back-team support) attitudes and behaviours around communicating, reporting, and recognising SRC, as well as acknowledging (the perceived ‘invisible injury’) concussion as an actual injury. PDs/PNs habitually implemented a range of nutritional strategies with their athletes. Some participants mentioned trialling or recommending nutritional protocols based on their personal experiences—either having personally sustained or having supported athletes with SRCs. These included carb-reloading by increasing carbohydrate-rich foods, reducing the consumption of omega-6 rich foods, promoting the intake of omega-3 fatty acids, and acutely supplementing omega-3s high in DHA and creatine and adding in a probiotic to support brain healing after a concussion.

### 4.1. Nutrition Protocols Being Implemented Irish PDs/PNs

Some participants did not have any knowledge of the role of nutrition in supporting an athlete’s recovery following a head impact and concussion diagnosis. PDs/PNs stated feeling unsure and ill-equipped and did not clearly understand their potential role as practitioners or how to implement nutrition in supporting concussion recovery. Many, in this instance, recommend nutrition strategies that adopted a comprehensive approach embodying athletes’ individual needs post-SRC without truly knowing or understanding the purpose behind them. Meanwhile, the strategies that PDs/PNs mentioned included being proactive, reaching out to athletes, and evaluating specific nutritional requirements following SRC to fuel and support their brain healing and recovery. Participants emphasised the importance of supporting athletes’ physical and mental health by ensuring nutrition support was tailored and did not strain their mental energy. Overall, PDs/PNs took a proactive and comprehensive approach when applying nutrition to help manage concussion recovery, resetting athletes’ habits and refocusing support to facilitate brain healing. Meanwhile, a range of acute nutritional strategies specific to support SRC were mentioned by PDs/PNs; these included supplementing with creatine monohydrate, consuming omega-3 fish oils rich in DHA, incorporating a probiotic, and replenishing carbohydrates through dietary means.

Creatine monohydrate supplementation was actively cited by participants (n = 11/17) and used to promote athletes’ strength and performance to be further incorporated and implemented in the event of a concussion. Using creatine acutely following an impact to saturate the brain has the potential to replenish levels by 3–10%, thereby restoring energy homeostasis and potentially blunting symptoms experienced by athletes [48,61,62]. The brain synthesises creatine endogenously; therefore, exogenous supplementation with the purpose of repleting levels may differ for individuals due to dietary preferences and age (> in older adults) [47,63]. Nonmeat or seafood eaters such as pescatarians, vegans, and vegetarians have a lower dietary intake of creatine and skeletal muscle stores, making them more responsive to supplementation than meat eaters (omnivores) [64]. Furthermore, during a period of neurometabolic stress, like after a concussive impact [47,63], brain tissue shows slower responsiveness and ability to replenish creatine levels [47,64] compared to muscle cells. Therefore, loading creatine to saturate brain cells may prove effective following SRC. However, it is crucial to determine a specific dosage tailored for injured brain tissue [47,62,65].

In a non-injured state, a creatine dose of 5 g (g) (approximately 0.3 g/kg body weight) four times daily (totalling approximately 20 g/day) over 5–7 days is suggested [66]. However, brain tissue may require higher doses (>20 g/day) for a more extended period to accumulate adequate levels [47]. Therefore, utilising creatine as a strategy like those employed for brain disease states, involving high doses administered over extended periods (e.g., 0.3–0.8 g/kg/day), may be more effective post-SRC [65].

Several participants mentioned prescribing 5 g of creatine daily to their athletes for muscle mass and performance (i.e., strength) properties. However, this strategy may not be sufficient following SRC and should be tailored to athletes’ creatine requirements (i.e., are they currently taking/not taking the supplement?) at the time of their impact. In combat sports, athletes must make weight to compete; this often includes cutting strategies that should be strategically planned and managed by PDs/PNs. Creatine promotes intramuscular water retention; therefore, to reduce weight (water), some athletes undergo a supplementation washout. Theoretically, these weight-cutting practices may make athletes more susceptible to concussion, especially if cuts are managed closely by a PD/PN. Dehydrated athletes who fail to rehydrate sufficiently post-weigh-in will be more susceptible to concussion, having reduced cognitive function and brain protection [67,68].

Further, in the absence of a PD/PN, athletes become more vulnerable as they may not receive creatine supplementation when a known concussion occurs. One participant [PN 12] (n = 1/17) recommended a creatine protocol involving a loading phase, which is being implemented with Gaelic athletes following SRC. The protocol included acutely increasing athletes’ daily creatine monohydrate dose from “5 g a day” to “20 g a day” [PN 12] following SRC (see Figure 3).

Presently, the optimal protocol for creatine supplementation in the context of SRC requires further investigation. While interval doses of 5 g, totalling 20 g daily for a 5 to 7-day period, are considered safe, these recommendations primarily support muscle tissue strength and performance capacities and may not be as effective in enhancing SRC recovery [48,69].

#### 4.1.1. Omega-3–Docosahexaenoic Acid (DHA)

Omega-3 fish oil supplementation was actively implemented by participants with athletes to support brain health, cognitive function [PN 2], healing, and tissue recovery [3,38]. Participants cited a general need to increase athletes’ intake of omega-3-type fatty acids using foods and supplements post-SRC without stating clear rationales or protocols. PN 6 clearly specified ensuring that athletes reduce their intake of foods high in omega-6 type fatty acids due to their pro-inflammatory properties, which may hinder brain recovery if prolonged [36].

In practice, athletes’ omega-3 intake is generally poor, and this is partly due to ‘Westernised’ dietary intakes being more abundant in convenience and processed foods rich in omega-6 fats (i.e., processed meats, fried and greasy foods, and vegetable oils). Consequently, athletes risk having pro-inflammatory brain environments deficient in DHA, with limited neuroprotection and worse adverse effects (i.e., symptomology, oxidative stress, inflammation), impacting their healing and recovery following concussion or RHIs [3,40].

One participant [PN 12] (n = 1/17) recommended an omega-3 supplementation strategy post-SRC that was implemented with Gaelic athletes. The aim was for athletes to consume a daily dose of 3 g of omega-3 to achieve a target ratio of 1.8 to 1.2 of EPA:DHA, aiming to increase brain DHA levels following a concussive impact (see Figure 3).

“*Ehmm, 3 about 3 g of, so that’s about 1.8 g. I think it’s like 1.8 to…the ratio was like 1.8 to 1.2, EPA:DHA…with the concussion. I’m trying to get the DHA levels up a little bit more…*”[PN 12]

However, further investigation (by E.F.) into the specific supplement and brand mentioned identified that the dosing cited (*1 cap = 500 mg EPA, 250 mg DHA/1000 mg omega-3 (PN 12 mentioned a 3 g dose, which equates to 3000 mg omega-3 = 1500 mg EPA and 750 mg DHA and 750 mg omega-3 filler–ratio 2:1:1)) may not constitute a realistic SRC recovery protocol.

Further research is needed, but current evidence supports PNs’ use of omega-3s, suggesting that high-dose DHA, either alone or combined with EPA, may optimise recovery and provide a protective nutritional foundation for the brain tissue after SRC. Several protocols are currently being tested to rebuild brain tissue and aid recovery following SRC. Dr Lewis recommends a three-step protocol utilising a high-quality 1000 mg omega-3 fish oil (EPA + DHA). The protocol involves (1) saturating the brain cells with a 9000 mg dose for 7 days, followed by (2) a 6000 mg dose for an additional 7 days, and (3) a maintenance dose of 3000 mg daily after that (protocol link: https://atlantictu-my.sharepoint.com/:b:/g/personal/emma_finnegan_atu_ie/EWWfBeX5lttCtsWGfSz2faMBH--eSJv2oNySP5afNaIWuw?e=SFfYpw (accessed on 25 June 2023), [69]). Falk [70] trialled a combined omega-3 protocol using a tiered dose of 6000 mg for 4 weeks and 1200 mg for 8 weeks (1000 mg (500 mg DHA and 100 mg EPA) capsule) and found it helped settle post-SRC symptoms (92.8%) [3,71]. Additionally, Miller [39] identified that a dose of 2000 mg/day of DHA is well tolerated by concussed athletes and quickly alleviates their symptoms [3,39]. Meanwhile, Heileson [40] further found that a dose of 2000 mg DHA in athletes in precompetitive and competitive seasons was effective at reducing head trauma biomarkers (serum Nf-L) associated with RHI.

#### 4.1.2. Carbohydrates

Acutely reloading carbohydrates post-concussion was mentioned by one participant [PN 2] (n = 1/17); this protocol is based on research suggesting that it may increase brain tissue energy availability effectively [59]. Carbohydrates are a preferred brain fuel source; therefore, consuming them will ensure a supply of glucose is available to enhance brain function and performance. In the case of SRC, the rationale is to provide glucose support to the brain, which may help alleviate symptoms related to energy availability discrepancies by delivering substrate to make ATP and fuel recovery. The protocol involved concussed athletes consuming a dose of 400–500 g of carbohydrate (via corn starch, using a 26 g dose) at controlled time points (a) within 30–60 min following diagnosis, then hourly (over 4 h), and then (b) twice daily for 5 days during their RTP [59]. PN 12 mentioned implementing aspects of Frakes’ [59] protocol in a rugby setting by ensuring athletes ate enough carbohydrate-rich foods acutely post-SRC. At present, the literature on the impact of acute carbohydrate administration for fuelling the acute hypermetabolic period following SRC is mixed and very limited [39]. Despite this, it is important to consider the quality of carbohydrate foods consumed as processed and refined types can have a less favourable impact on healing and recovery from concussion.

#### 4.1.3. Probiotics

One participant [PN 11] (n = 1/17) highlighted the significance of incorporating a probiotic supplement after SRC following personal experience (see Figure 3), having sustained three consecutive impacts within 18 months. PN [11] shared that including a probiotic in their recovery plan, along with maintaining regular eating intervals, had helped support their gut function. SRC is a neurological injury that can result in symptoms like vomiting, which are triggered by the mismatched signals between the brain’s vagal nerves and the stomach. This mismatch leads to an inflammatory response that disrupts gut function, affecting food transit, metabolism, and nutrient absorption. Consequently, it can disrupt the gut microbiome’s balance, leading to dysbiosis and prolonging inflammation and recovery [72,73]. Pilot findings have confirmed that the changes that occur in an athlete’s gut microbiome following SRC do negatively influence performance and prolong inflammation [74].

Therefore, restoring the microbiome environment of an injured athlete may positively interrupt the gut response, mitigate symptoms, and ease neurological deficits following SRC. Therefore, probiotics have the potential to positively alter the gut microbiome and support communication between the gut and the brain. This may help reduce athletes’ symptoms and inflammation and improve their digestion. As a result, nutrient absorption and energy availability to the brain may be facilitated, as reported anecdotally by PN [11] (see Figure 3), to support SRC healing and recovery.

### 4.2. Impact of Back-Team Collaboration and Communication

#### 4.2.1. Communication, Collaboration, and SRC Reporting

Effective communication and collaboration amongst backroom team practitioners and athletes are key mediators in establishing effective recovery strategies after injury diagnosis [17,75], especially after SRC. PDs/PNs know that differing levels of communication exist as part of working in a team and recognise that a “*decent backroom team*” [PN 5] needs internal collaboration to formulate and deliver suitable protocols to manage SRC recovery. Participants highlighted the importance of having a well-coordinated, athlete-centric backroom team defined by shared values, professional respect, and effective communication. This cohesive backroom teamwork ensures the prompt and effective implementation of protocols and support for athletes when sidelined due to injury. Open channels of communication promote collaboration and enable the sharing of insights from multiple perspectives to make informed athlete-centred decisions, such as athletes’ RTP strategies. On the other hand, when communication breaks down, it can lead to increased misunderstandings and a risk of one-sided decision making [17,75]. The limited recognition of PDs’/PNs’ role by other team members in the event of injury has led to incidents where participants experienced closed communication channels, impacting their awareness of their athlete’s history with SRC. Instead, each backroom practitioner should understand and endorse a multifaceted approach when supporting athletes, with clarity on their role and one another’s contribution within that following SRC diagnosis. Otherwise, a lack of awareness and respect among backroom practitioners and misaligned values may compromise the quality of care provided to injured athletes [75,76]. PDs’/PNs’ awareness of SRC incidences is shaped by the culture and reporting practices within their sports setup. This was evident by the limited awareness amongst participants regarding athletes’ SRC experiences, highlighting the crucial role of collaboration and open communication channels. Interestingly, those involved in rugby or elite-level sports exhibited greater awareness of SRC than those in Gaelic, hockey, combat, or amateur-level sports. Additionally, symptoms of SRC are not externally visible and are solely experienced by athletes, which increases the risk of underreporting, injury masking, and underestimating its severity due to insufficient knowledge. Nonetheless, practitioners on the sideline rely heavily on athletes’ self-reporting of symptoms [77,78]. Poor communication among backroom team practitioners, combined with these factors, may be the underlying cause of PDs’/PNs’ limited understanding of their role and responsibilities in facilitating athletes’ SRC recovery.

#### 4.2.2. Roles and Perceived Responsibility

The attitudes of athletes and backroom team practitioners toward concussion, along with the subjective nature of the injury, contribute to underreporting and shape the perceptions of PDs/PNs regarding their role, influencing recovery outcomes. PDs/PNs involvement in a team or individual athlete support setting goes beyond a tick-the-box exercise and is a vital component of the team and athletes’ backroom support. Their role extends not only to situations where there is a perceived need or when an athlete’s performance is a priority but also when physical and invisible injuries occur. Athletes require individual support for tissue healing and rehabilitation. This cannot happen if a PD’s/PN’s role is not valued but seen as a nonchalant tick-the-positive nutrition box exercise or a lower priority and sometimes cancelled to save funds. Coaches, physios, medics, and athletes should be aware that the role of the PD/PN goes beyond superficial support and requires involvement during all aspects of communication around athletes’ acute health, performance, and injury outcomes. All members have an active duty of care with a common goal to support and prevent a breakdown in all pillars of health on the athlete’s and teams’ path to their goals and recovery.

### 4.3. Limitations

The findings of this research reflect a particular period, and it is important to acknowledge that performance and injury nutrition are rapidly evolving fields. Therefore, the factors highlighted in this study may not remain applicable in the future. Despite the relatively small Irish cohort of PDs/PNs (n = 17), this study has provided valuable insights into areas that require consideration by sport performance dietitians, nutritionists, coaches, and other team professionals in the context of managing SRC. The research has identified current best-practice nutritional strategies and potential protocols to implement and support athletes’ SRC recovery, acknowledging the clear gaps in research and their impact on PD/PN decisions and their implementation of SRC management protocols with athletes. It is important to note that PDs/PNs often have limited or no concussion education. Participants demonstrated varying levels of knowledge regarding concussion as an injury and the potential role of nutrition as a management strategy. The sporting environment within the team and professional practitioner setup significantly influenced participants’ awareness, diagnosis, and management of athletes’ SRC. Moreover, specific attitudes and norms related to sporting culture were observed across different sports, levels, and team environments. In addition, the availability of support resources for athletes was strongly influenced by the level of sport at play (elite vs. amateur) and the teams’ gender (female vs. male), which impacted PDs’/PNs’ awareness and the level of support provided to athletes following SRC.

### 4.4. Future Recommendations

1.Further research that is robust and can be safely implemented to support and manage concussion recovery in a sporting context is needed. Irish PDs/PNs have cited using creatine, omega-3 fatty acids, probiotics, and carbohydrate reloading strategies. The availability of human-based evidence is limited, highlighting the necessity for additional investigation and research in this area.2.Concussion and head injury education and training should be mandatory for all healthcare practitioners supporting athletes, including nutrition experts (PDs and PNs), psychologists, and physical therapists (physio, rehabilitation, and athletic) working with or as part of the back team. According to the latest concussion management protocols, a diagnosis should ideally be conducted by medical healthcare professionals such as a doctor or physiotherapist [1], which is not feasible for many sports settings (i.e., amateur, underage, and female). To address this limitation, we recommended that all healthcare practitioners in the multidisciplinary care team receive comprehensive training on brain injuries and concussions.3.Open and transparent communication among all back-team members, actively involving PDs/PNs in player analysis, performance, and injury discussions, is needed. This collaborative approach will help mitigate the risk of missing acute injury management and the subsequent risk of poor recovery outcomes for athletes.4.Regular monitoring of athletes’ hydration and mental capacity status should be implemented during pre-match hydration test protocols to identify and prevent cognitive decline and, as a result, help reduce the risk of an impact occurring during gameplay. PDs/PNs can use the CRT6 and SCAT6 tools to educate athletes about the effects of their hydration and its impact on brain function. They can use it as evidence to address suboptimal nutrition, assess performance, and measure the recovery of athletes with SRC. By conducting regular evaluations and the monitoring of athletes, PDs/PNs can enhance nutrition protocols for concussion recovery and support athletes, ultimately helping them prevent injury and achieve optimal recovery and RTP outcomes.

## 5. Conclusions

PDs’/PNs’ use of nutrition protocols with athletes following acute SRC diagnosis was linked to their knowledge and the scientific evidence available. Nutrition implementation, as a result, may be overlooked or implemented with uncertainty due to limited evidence, which could negatively affect the potential for optimal SRC recovery outcomes. From PD/PN practices, it is evident that there is a need for research to uncover safe and practical nutritional protocols that they can use to mitigate the effects of SRC and support athletes’ recovery and return to play.

## Figures and Tables

**Figure 1 nutrients-16-00497-f001:**
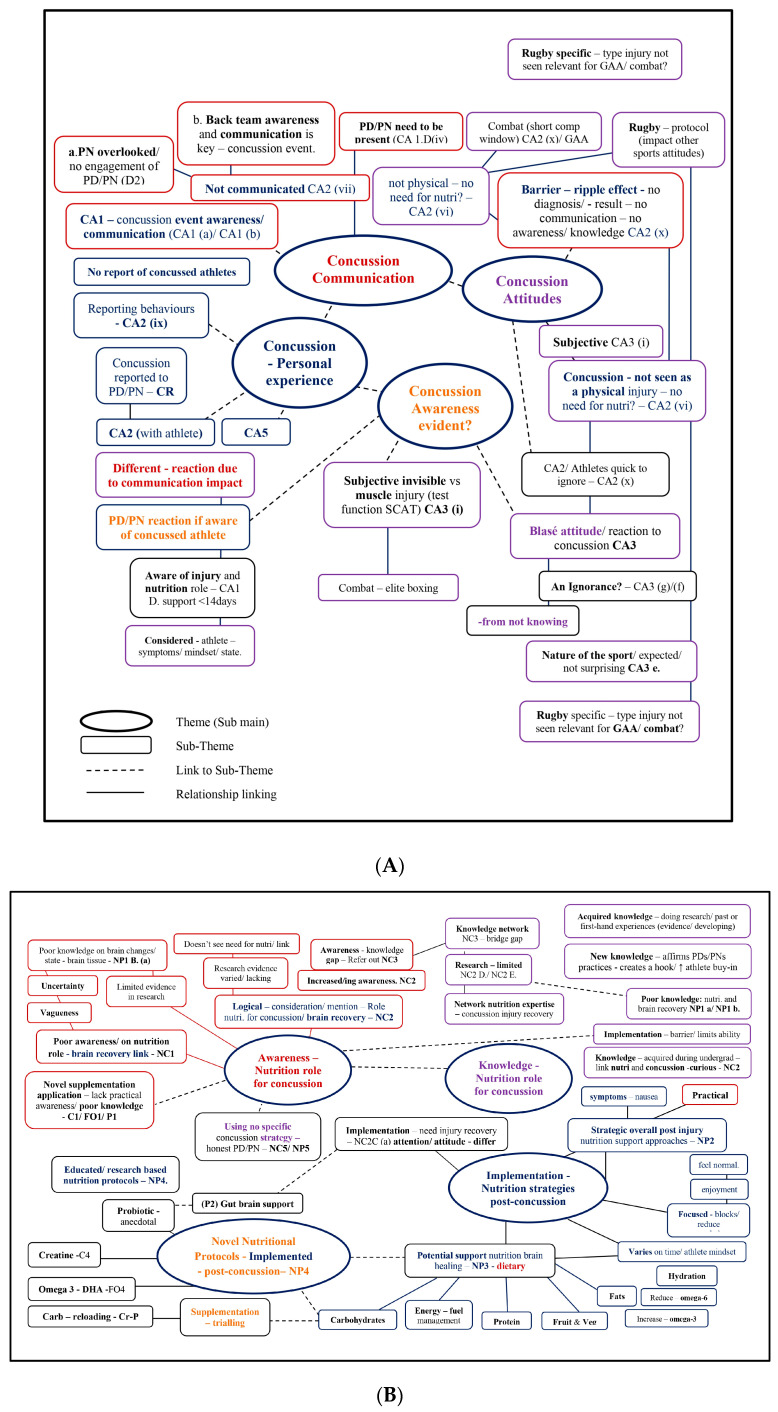
(**A**) Reflexive Thematic Analysis (Phase 5). Awareness of concussion—thematic map demonstrating four subthemes (s-T). (**B**) Reflexive Thematic Analysis (Phase 5). Nutritional support for concussion management—thematic map demonstrating four subthemes (s-T).

**Figure 2 nutrients-16-00497-f002:**
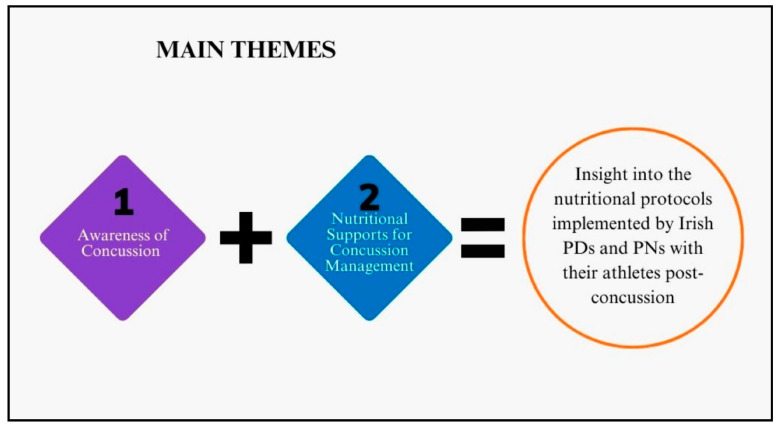
Reflexive thematic analysis. Result: main themes addressing the research question. Note: PD(s): Performance Dietitian(s), PN(s): Performance Nutritionist(s).

**Figure 3 nutrients-16-00497-f003:**
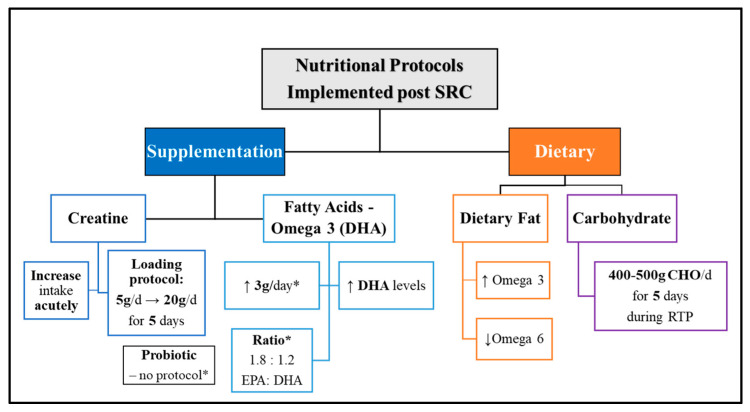
Novel nutritional protocols implemented post-SRC by PD/PN participants. Abbreviations—DHA: Docosahexaenoic acid, EPA: Eicosapentaenoic acid, CHO: Carbohydrate; RTP: Return to Play/Practice. Symbols: ↓: Reduce, ↑: Increase, g: grams, d: day, *: note.

**Table 1 nutrients-16-00497-t001:** Participant profiles and characteristics: PD/PN background, education, roles, employment, and athlete types.

Participants Profile											
Total Sample (n)					n = 17						
Gender					Male				Female		
% (n)					59% (10)				51% (7)		
Undergraduate degree		BSc Dietetics and nutrition sciences	BSc Human nutrition sciences	BSc Nutrition and sports sciences		BSc Health, sport, and exercise sciences		BA Education with science			Other (BA, BCom, BBS non-science degrees)
% (n)		11.8% (2)	17.6% (3)	5.9% (1)		35.3% (6)		11.8% (2)			17.6% (3)
Post Graduate degree (MSc/PhD)		MSc in Sports Nutrition	Dip Sports Nutrition	MSc Sports and Exercise Nutrition		MSc in Human or/and Nutrition,		MRes in Sport and Exercise Nutrition			Doctor of Philosophy (PhD)
(n)		(4)	(3)	(9)		(2)		(1)			(5) Specialisms included: Energy, Sports, and Exercise Nutrition Bone Health, Exercise Biochemistry, Bioactives in Health and Exercise.
SENr Registered % (n)		Yes 100% (17)									
Previously concussed (history) % (n)		Yes 23.5% (4)				No 76.5% (13)					

**PN-ID**	**Type Athlete/Client**						**Teams/Organisations**			**Part/Full-time**	
PN 1	Elite teams: male senior and U20s Gaelic footballers—county-level.						GAA			Part-time/contract roles	
PN 2	Elite teams: male athletes in rugby union—academy and senior squads; male Gaelic footballers—county-level.						IRFU, GAA			Full-time	
PN 3	Elite individual athletes (mixed); elite teams: rugby union (senior, academy males/7s, XVs female) and senior (male) Gaelic football and hurling athletes—county and intercounty levels.						Sport Ireland, University scholarships, Athletics Ireland, IRFU, LGFA, GAA			Part-time/contract roles	
PN 4	Elite teams: men’s hurling and ladies’ and men’s Gaelic football (male/female) athletes. Individuals: amateur/non-elite-MMA, soccer athletes.						LGFA, GAA			Part-time/contract roles	
PN 5	Elite amateur teams: Gaelic football athletes; individual: amateur/non-elite endurance and rugby athletes						GAA			Full-time	
PN 6	Individual: retired professional/high-level amateur-endurance (runners, cyclists), kickboxing, high-intensity functional training, powerlifting athletes						N/A			Full-time-self-employed	
PN 7	High-level amateur teams: U20 hurlers (county) and senior men’s Gaelic footballers (intercounty). Individual: soccer athletes.						GAA			Part-time roles	
PN 8	Elite/Professional teams: academy, youth, and senior men’s rugby union (current). Women’s soccer, senior ladies’, and U20s men’s Gaelic football and hurling at county and intercounty levels (past). Individual: Triathletes (paralympic squad, past).						IRFU, LGFA, GAA, Women’s FC			Full-time	
PN 9	Elite: Women’s XV and 7s and men’s 7s academy and sub-academy rugby union teams. Amateur: junior Gaelic intercounty team athletes.						IRFU, GAA			Full-time	
PN 10	Teams: U21s and mixed 1–1 Gaelic footballs. Individual: professional and amateur combat male/female athletes—boxers, MMA, and kickboxing.						GAA			Full-time-self-employed	
PN 11	Teams: Elite/high-level women’s soccer, hockey, and rugby union squads. Individual: mixed 1–1 athletes.						WFC, women’s hockey, IRFU, SN private practice			Part-time/contract roles	
PN 12	Elite/high-level amateur teams: U20s and senior Gaelic footballers and hurlers.						GAA			Part-time/contract roles	
PN 13	Elite teams: women’s hockey, individual junior gymnastic athletes, and endurance athletes (cyclists).						Sport Ireland			Full-time	
PN 14	Teams: professional rugby union and high-level amateur Gaelic football athletes. Individual: elite/professional Olympic boxers, rowers, athletics, judo, and cricket athletes.						Sport Ireland			Full-time	
PN 15	Teams: senior and U21 Gaelic footballers and hurlers, academy male rugby union athletes. Individual: Gaelic footballers, hurlers, and jockeys.						GAA, IRFU			Part-time roles	
PN 16	Team: rugby/soccer/hockey/Gaelic hurling, camogie, football, club, intercounty, ladies, men’s athletes. Individual: competitive/high level/weight class athletes/HIFT/multi-sport athletes (boxers, Olympic, power weightlifting/MMA)/endurance athletes.						N/A			Full-time	
PN 17	Teams: professional rugby union—male academy/NTS squads (16 years up), high-level amateur intercounty Gaelic football and Camogie athletes. Individual: endurance athletes.						IRFU			Part-time roles	

Abbreviation notes: SENr: Sport Exercise Nutrition register; BDA: The British Dietetic Association; PN: Precision Nutrition; ISAK: The International Society for the Advancement of Kinanthropometry; NTOI: Nutritional Therapists of Ireland; IRFU: Irish Rugby Football Union; FC: Football Club; GAA: Gaelic Athletic Association, MMA: Mixed Martial Arts, HIFT: high-intensity functional training, SN: Sports nutrition.

**Table 2 nutrients-16-00497-t002:** Theme 1: Irish performance dietitians and nutritionists’ awareness of concussion. Developed themes, subthemes, sample codes, and example quotes.

Theme	Subtheme	Example Quote
1. PD/PN Personal experiences with concussion injuries	Concussion diagnosis reported to PD/PN	…your first most common will be, ehmm, connective tissue injuries. So, like your shoulders…shoulders, ACLs, ankles syndesmosis, then followed by concussion, and then you have your hamstrings and calves like soft tissues, and then you have your broken bones, and then your kind of back injuries would be. [PN 9]
	Concussions not reported by athletes to PD/PN	…you never…never even hear of like concussions reported back to you, but God knows, even though it’s probably likely you’ll never hear back! [PN 12]
	Personal concussion experience	First impact: …the first one happened during a hockey match… I just stepped back, and the player behind me…stepped back, and it was just like totally innocuous. No big deal; we didn’t even hit heads that that hard! I played on…totally, sort of out of it! I went off, and then I tried to insist on going back on, but then eventually the penny dropped that I wasn’t ok and like I couldn’t keep my eyes open… I felt fine again a few hours later… If I’m honest, I didn’t change anything about the way I was eating, then. I just…sort of slept a bit more. [PN 11] Second impact: …next one, then only about four months later, I got a ball right into the face; a hockey ball. I kind of knew then I’m obviously slower… I did make the change (nutritionally) at that stage…I would be, you know, eating more regularly…making sure my protein intake would better across the day…the probiotics… I was a bit worried. [PN 11] Third impact: …the third one. I was playing with my kids in the swimming pool…managed to smack my head off the bottom of it…then…I didn’t play hockey for quite a while because I wanted…to get a brain scan… Three in 18 months is a lot! [PN 11]
2. Concussion injury communication	Back-team communication key-acute injury	…I know when there is a decent backroom team that are all together, and I know when… they don’t really believe in all the different pillars as much as their own pillar… If you’ve got a backroom team that is set up in a very efficient way, it’s just then firing off a message to one person who does their part… I would work very closely with one set of backroom teams, the doctor and the physio, and the S&C coach… Even though they’ve changed… the doctor never has… there has always been a kind of a multi-faceted approach… Whereas in other situations, it hasn’t been as strong, you know, and I think that makes a big difference in terms of catching the acute phase. [PN 4]
	Open Communication	So, making contact… as soon as possible, but again… if you know the doc is on to them if you know the coach, the manager is on to them… if you know the physios are on to them, it is sometimes best to void your time… I’ve gotten a little bit better at identifying when are you helpful and when are you just annoying them you know? [PN 5]
3. Poor Concussion Reporting-Risk	Lucky or poor communication and reporting?	I’m…I’m so I’m so lucky that in just having the conversation with you there now… I’m just thinking back like five years, five years working in performance nutrition directly, and I have never had anyone with real…real acute injuries like. [PN 10]
	Concussion underdiagnosed or a reality of poor communication?	So, ehhh…luckily, we don’t really get too many of them; now, like that, I can; I can miss some as well just by not being on site too often. Ehmm…again, like I’ll go back to my old role of just turning up when, ehhh…when the, the rugby season is a little bit over. [PN 4]
4. Concussion Attitudes and Behaviours	Reaction to injury frequency of athletes-nature of sport?	Overall, there’s nearly always, have some kind of knock or a niggle or a strain… When we’re coming out off the back of matches ehmm to be honestly, injuries…they’re very varied… I’m always just always still, even in shock with these guys! Like, how can you cope with it like they’re just constantly in a state of basically recovery because they come out so beat and bruised from games. [PN 8]
	Athletes blasé risky attitude–concussion	Oh yeah! I think they are very quick to dismiss…injuries, especially when, I suppose, there’s no, not that there’s no obvious symptoms of it. But you know, they’re not carrying their leg, for instance, or, you know? So, I definitely think they will try and mask it… I think your senior, more experienced players don’t do that as much, but I definitely think your younger players will do what they can to try and get back out… [PN 7]
5. Concussion Awareness Evident	Elite-level subjective injury ‘carrot’ proof protocol	…boxers when they’re dehydrated, looking at the SCAT test. So, we’ve actually used that as part of our showing them… Showing them that if they’re dehydrated, their response to the SCAT is actually a lot less. So, you know, we’ve…put that in place some years ago around, you know, trying to educate them, to show them… That if you cut back on your fluids or you dry out too quickly…the reason that they may be drying out too quickly is because they’re not following a better weight-making process. [PN 14]

Abbreviation notes: PD: performance dietitian, PN: Performance nutritionist, E.F.: interviewer.

**Table 3 nutrients-16-00497-t003:** Theme 2: Nutritional support for concussion management mentioned by Irish performance dietitians and nutritionists. Developed themes, subthemes, sample codes, and example quotes.

Theme	Subtheme	Example Quote
1. Awareness of Nutrition in Concussion Management	Poor awareness	Not too sure, really, just keep up the good habits. But I haven’t looked at it… Is there much? Is there much research on concussion and nutrition immediate or…? [PN 1]
	Aware with limited knowledge	Yeah…yeah, I look; I’d be the first admit now I don’t… I know “X” and a few lads up in “x” are really interested in this area, and it’s it is really interesting! So, I’m kind of keeping an eye on creatine, fish oil supplementation, ehmm, and just to see the effects on recovery… If there’s anything there with creatine and fish oils. [PN 5]
2. Knowledge on the role of nutrition in concussion recovery	Acquired knowledge	Yeah, it’s; it’s an area I’m doing a bit of reading around at moment. But definitely, concussions are not that well-researched. But I think from what I have read, there’s a couple things you can do, and creatine, I believe, is important in the…not even the acute phase of, of, of having a concussion, but I think consistently for optimal cognition. [PN 2]
	Personal led knowledge	…what would I implement with somebody else? I, I think, to be honest…the rest and the regular eating are…crucial and then… Whether, whether the extra probiotics or… Do you know whether they make that much of a difference? I don’t really know… But the anti-inflammatory, the protein, the carb, I think they would be crucial. Obviously, you know…you know your…your healthy fats as well. But the diet…the diet itself is probably the most important thing. [PN 11]
	Nutrition knowledge network	With regards to concussion, then we have two dietitians. One that actually has worked in rugby and who would I suppose work and are more familiar with the concussion side of things… [PN 13]
	Expertise in nutrition risk	…if you look at concussions, like concussion, is like shortage of energy in the brain at the time of impact, so…you could, like. I’d broadly say it to people, you know…’just to make sure you eat enough during this time, and kind of, you know’, were kind of very basic interventions, but nothing, I don’t think we ever…got to a stage where we were comfortable giving nutrition advice to someone who was concussed because there was stuff out there on creatine and stuff on there, high dose DHA, but we never…never formalised that to a policy. [PN 9]
3. Implementation—Nutrition post-concussion	Implementation with uncertainty	Ehmm, so I’m kind of keeping an eye on creatine, fish oil supplementation…and just to see the effects on recovery…if there’s anything there with creatine and fish oils. So, I would be kind of promoting or recommending to just keep those, the dosage on those… ehmm, keep including those if the lads have a concussion, and I would be in touch with the lads as well, like… …focus on the hydration. Keep on top of the omegas and the creatine. I’m not fully sure about the creatine yet, but they’re taking it anyway for other reasons and I think there could be? [PN 5]
	Focused energy/symptom management	…one of the players who had been concussed had two in four weeks and was, you know, quite nauseous, wasn’t able to eat a lot. So, it was more around I suppose having smaller meals, more frequently liquid, a lot of liquid meals and stuff… I just didn’t really feel like it was settling him, but obviously, we are still, I suppose, were trying to get back to where he needs to be and be ready for when they get the go ahead to go back and train… So tried to ensure energy levels were kept up just through kind of liquid-based meals, small little snacks, nothing that was going to cause… stomach aggravation… So, it was more kind of in that, you know, couple days after… after the knock as opposed to long term [PN 7]
	Strategically blocked collaborative athlete support	I try to do it in smaller focuses instead of bombarding them all at once… So, let’s say they initially we try to break down even with the physios; we’d break down their rehab into, like, phases… So, the initial being, their protection phase, for example… I try row in with that being with ‘this is your initial focus’ instead of trying to bombard them with loads. Saying, ‘this is what I really want to get out of this block’, and then we tried to do it in focus. [PN 8]
	Supplementation–refocus	…if they do get a concussion, I kind of re-emphasise that. I’ll be like, you know… for some, not everyone will take fish oils ehmm, even though they have done… So, I’ll re-emphasise that you should be taking fish oils, and I’ll even tell them to load creatine and during that period… [PN 12]
	Dietary fat intake	your brain tissues are very, very reliant on kind of fat as its main fuel source… in terms of kind of recovery… So, a big, big focus on kind of their healthy omega 3s, and a reduction in obviously their pro-inflammatory omega 6s [PN 6]
4. Potential barrier	1. athletes’ attitudes	I would say that the physical ones… they are probably more… (shaking head) open to the fact that nutrition can play (emphasised tone) a role immediately… Whereas the likes of the concussions, as you said, like it’s not visible… So maybe a little bit more towards the towards the physical ones… [PN 17]
	2. athletes’ nutrition adherence	…they are obviously fine, but like they… Well, see, I don’t know if that they are actually taken it! But the guys who have been concussed recently… I definitely know one of them 100% was because he was looking for more creatine and everything, you know because he was loading up on it… [PN 12]
5. Novel nutrition protocol implementation post-concussion	Acute creatine intake	Again, I’ve done a little bit of research into it, and there is a small bit around concussion and creatine. [PN 8] Yeah. [E.F.] So, a lot of our athletes would take creatine. But if they haven’t or around a concussion, potentially increasing that creatine intake, ehmm… In that period of concussion, to help with that energy availability side of things. Ehmm, if the brain can’t fully use their glucose efficiently. So, I have trialled that; again, there was a little bit of research around it. But ehmm, aside from that, no there is nothing else. [PN 8]
	Carbohydrate-energy availability	Matthew Frakes is a researcher in Pittsburgh. I think he spoke about the energy availability and that like they compared guys that I don’t know-was it like 4 or 500 g of carbs in the 5 days, after concussion versus the guys that had a low carbohydrate diet and the symptoms for the guys that had more carbohydrates were got back to baseline that bit quicker. So, that’s kind of something I have been doing; it’s like, well, if a guy is in a gradual return to play protocol, then can you eat enough food? [PN 2]

Abbreviation notes: PN: Performance Nutritionist, PD: Performance Dietitian, E.F.: Interviewer.

## Data Availability

The original data, unfortunately, are unavailable to protect participant confidentiality. However, the data presented in the study article are included in the Appendix A. Further inquiries can be directed to the corresponding author.

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
