# Peer review of "Nutritional Considerations of Irish Performance Dietitians and Nutritionists in Concussion Injury Management"

_nutrients, 2024, doi:10.3390/nu16040497_

Round 1
Reviewer 1 Report
Comments and Suggestions for Authors
Overall, the study by Finnegan and colleagues highlights an important topic for those practitioners caring for athletes with traumatic brain injury (TBI). The manuscript is well writing, with clear rationale, methods and outcomes. The discussion addresses the important outcomes related to the major semantic themes derived from the study, specifically awareness of concussion events and use of nutritional supports for concussion management. With the primary focus on those nutritional protocols that yield positive outcomes for patient care post TBI. Overall, the recommendations are very well-considered.
Just one major comment that the authors may wish to consider. Is a n=17 sufficient sampling? While I agree that saturation would likely occur with this sample size if the PNs were sufficiently homogenous, however it seems as though there are very different PNs experiences out there dependent on the networks through which they were sought and this clearly influenced the first of the themes, awareness. There is a bias that those associated with IRFU as they are more aware of TBI in general and more likely to include it as standard of care question when considering the history of patients and their subsequent needs for recovery. This is reflected in the discussion, and it begs the question of whether a sampling of more PNs from other non-IRFU networks would yield more thematic diversity for the second major theme of nutritional supports.
Author Response
Response: We thank you for reviewing our work thoroughly and providing positive commentary on the manuscript.
- In relation to sampling, the Sport and Exercise Nutrition register (SENr) and sporting body networks across Ireland were used to identify potential participants, yielding 57 potential participants; 48 of the 57 were invited via email as they met the inclusion criteria and practised in Ireland. Following this invitation, 17 of the 48 invited confirmed they were available to attend an interview; 7 out of the 17 participants had experience working with the IRFU as a professional sporting body, while 10 did not.
- Your comment regarding sampling and a network biased towards rugby (IRFU) has been taken on board. However, the pool of performance nutritionists and dietitians (PNs, PDs) in Ireland is relatively small as outlined above, and there are limited full-time positions. In addition, many, if not all performance dietitians and nutritionists, will have some level of experience working with a high-level professional team that typically would have a full-time on-site performance dietitian/ nutritionist working with athletes; in Ireland, the only professional sport is rugby, whereas in the UK for example, this would include rugby or soccer. Therefore, in our opinion it is not a bias that the PD/PNs interviewed have rugby experiences. Furthermore, as highlighted by our study, even with their experience many did not have any knowledge of concussion or mTBI and it is not to be assumed that all those working in rugby have this knowledge.
Reviewer 2 Report
Comments and Suggestions for Authors
Thank you for submitting the manuscript “Nutritional Considerations of Irish Performance Dietitians and Nutritionists in Concussion Injury Management” to Nutrients. Overall, the study appears to have been well conducted, but I believe there are some areas for improvement. Furthermore, the fact that the manuscript is a little confusing is partly due to its length and, in addition, it has supplementary files.
- abstract: abbreviations must be defined the first time they appear. Consider including the n sample of the research.
- Regarding the sample, it seemed small to me for an interview survey. How much is the survey n related to the number of nutritionists or dietitians registered for the survey site?
- the introduction is a little out of focus. Maybe it's because it's long. All the issues highlighted in the introduction are justifications for carrying out the present study, but I believe that the justification for carrying it out is despite the lack of standardization of treatments by health professionals. Furthermore, the introduction is a good place to define the difference between nutritionist and dietitian.
- Line#154 and others: every time an abbreviation appears for the first time it must include its definition. Please check the entire text.
- Consider changing the connections between the balloons in Figure 2 with arrows to direct the reader as to the order in which the study was conducted.
- Line#311-315: the sentence does not seem complete.
- Table 2: one of the ethical principles of the research is that the publication of the results does not allow the research participant to be recognized. The way this Table is, this could happen. I believe that the best option would be to summarize this data in graphs using a relative percentage considering the study's audience.
- The results item has an excessive display of results that can be grouped in percentages, in figures or that do not need to be presented because although they were obtained by the study, they are not important for the study. It is important to emphasize that a manuscript is not simply a research report and therefore, much information needs to be rewritten in a more summarized form.
- Line#749: what does holistic approach refer to?
Comments on the Quality of English LanguageThe English needs minor adjustments for spelling errors, but the text needs adjustments for clarity and better readability.
Author Response
Comments and Suggestions for Authors: Thank you for submitting the manuscript “Nutritional Considerations of Irish Performance Dietitians and Nutritionists in Concussion Injury Management” to Nutrients. Overall, the study appears to have been well conducted, but I believe there are some areas for improvement. Furthermore, the fact that the manuscript is a little confusing is partly due to its length and, in addition, it has supplementary files.
- Authors' response: We thank you for your feedback on the manuscript and insightful comments on areas of improvement to our work thus far. Your advice and suggestions have been carefully considered and taken on board, prompting us to question certain sections and enhance overall clarity throughout the manuscript.
Abstract: abbreviations must be defined the first time they appear. Consider including the n sample of the research.
- Author changes: SRC sport related concussion(s) and carb to – carbohydrate, addition of sample seventeen; (n = 17).
Regarding the sample, it seemed small to me for an interview survey. How much is the survey n related to the number of nutritionists or dietitians registered for the survey site?
- Author response: In relation to the study design, 1-1 interviews were conducted with participants, it was not a survey analysis. The qualitative data were analysed using Braun and Clarke's thematic method. The potential sampling pool of performance dietitians and nutritionists working in Ireland is relatively small. Participants were recruited using the Sport and Exercise Nutrition register (SENr) and sporting body networks across Ireland, yielding 57 potential participants; 48 were invited via email as they met the inclusion criteria and practised in Ireland. Following the invitation, 17 were available to interview.
The introduction is a little out of focus. Maybe it's because it's long. All the issues highlighted in the introduction are justifications for carrying out the present study, but I believe that the justification for carrying it out is despite the lack of standardization of treatments by health professionals. Furthermore, the introduction is a good place to define the difference between a nutritionist and a dietitian.
- Author response: Amendments have been made to the introduction.
Line#154 and others: every time an abbreviation appears for the first time, it must include its definition. Please check the entire text.
- Author response Line #154 – amended; however, a full definition was included in the manuscript “COnsolidated Criteria for REporting Qualitative research (COREQ) (51).”
Consider changing the connections between the balloons in Figure 2 with arrows to direct the reader as to the order in which the study was conducted.
- Author response: Thank you for the recommendations for Figures 2a and b. We have not added arrows, as the - - - - - signifies each sub-theme linkage and each theme (T1/T2) has its own figure A/B .
Line#311-315: the sentence does not seem complete.
- Author response: This has now been edited as follows ‘Individual sports were also represented, such as combat sports, boxing, mixed martial arts, kickboxing, and judo (amateur and professional/elite levels), gymnastics (elite), athletics, running and cycling (elite, ex-elite, and amateur levels), rowing, endurance events; triathlon (Ironman) and competitive multisport, Olympic lifting, and powerlifting.
‘
Table 2: one of the ethical principles of the research is that the publication of the results does not allow the research participant to be recognized. The way this Table is, this could happen. I believe that the best option would be to summarize this data in graphs using a relative percentage considering the study's audience. The results item has an excessive display of results that can be grouped in percentages, in figures or that do not need to be presented because although they were obtained by the study, they are not important for the study. It is important to emphasize that a manuscript is not simply a research report, and therefore, much information needs to be rewritten in a more summarized form.
- Author response: Thank you for your manuscript amendment recommendations, as this will provide a lot more clarity and limit confusion. Changes have been made throughout the manuscript to increase readability, Table 2 has been summarised further to prevent participant recognition and Figure 2a, and b has also been amended for more ease in the flow of the thematic analysis for the reader.
Line#749: what does holistic approach refer to?
- Author response: Thank you for this comment, for clarity this has now been changed to a ‘comprehensive’ approach rather than holistic.
Round 2
Reviewer 2 Report
Comments and Suggestions for Authors
Dear editor,
Great efforts were made by the authors to answer the questions raised by this reviewer and to incorporate the suggested corrections. Therefore, I believe that the manuscript has greatly improved in quality and can be accepted for publication in this important journal.